# ARID1A loss is associated with increased NRF2 signaling in human head and neck squamous cell carcinomas

Vinh Nguyen[1,2], Travis P. Schrank[1,3], Michael B. Major[1,4]*, Bernard E. Weissman [1,2,5]*

**1** Lineberger Comprehensive Cancer Center, University of North Carolina at Chapel Hill School of Medicine, Chapel Hill, North Carolina, United States of America, **2** Curriculum in Toxicology and Environmental Medicine, University of North Carolina, Chapel Hill, North Carolina, United States of America, **3** Department of Otolaryngology, University of North Carolina, Chapel Hill, North Carolina, United States of America, **4** Department of Cell Biology and Physiology, Washington University in St. Louis, St. Louis, Missouri, United States of America, **5** Department of Pathology and Laboratory Medicine, University of North Carolina at Chapel Hill School of Medicine, Chapel Hill, North Carolina, United States of America

* bmajor@wustl.edu (MBM); weissman@med.unc.edu (BEW)

**Data Availability Statement:** All relevant data are within the paper and its Supporting Information files.

## Abstract

Prior to the next generation sequencing and characterization of the tumor genome landscape, mutations in the SWI/SNF chromatin remodeling complex and the KEAP1-NRF2 signaling pathway were underappreciated. While these two classes of mutations appeared to independently contribute to tumor development, recent reports have demonstrated a mechanistic link between these two regulatory mechanisms in specific cancer types and cell models. In this work, we expand upon these data by exploring the relationship between mutations in BAF and PBAF subunits of the SWI/SNF complex and activation of NRF2 signal transduction across many cancer types. ARID1A/B mutations were strongly associated with NRF2 transcriptional activity in head and neck squamous carcinomas (HNSC). Many additional tumor types showed significant association between NRF2 signaling and mutation of specific components of the SWI/SNF complex. Different effects of BAF and PBAF mutations on the polarity of NRF2 signaling were observed. Overall, our results support a context-dependent functional link between SWI/SNF and NRF2 mutations across human cancers and implicate ARID1A inactivation in HPV-negative HNSC in promoting tumor progression and survival through activation of the KEAP1-NRF2 signaling pathway. The tumor-specific effects of these mutations open a new area of study for how mutations in the KEAP1-NRF2 pathway and the SWI/SNF complex contribute to cancer.

## Introduction

Head and neck squamous cell carcinoma (HNSC) is the sixth most common cancer worldwide, with an anticipated 30% increased incidence by 2030 [1]. The 5-year survival for HNSC has improved modestly over the past three decades, from 55% in 1992–1996 to 67% in 2012–2017 [2]. Approximately 40% of HNSC are positive for human papilloma virus (HPV+), which

**Funding:** This work was supported by the National Cancer Institute [R01 CA216051] (MBM & BEW), the National Institute of Dental and Craniofacial Research [K08 DE029241] (TS) and the National Institute of Environmental Health Sciences [T32 ES007126] (VN). The funders of this study had no role in study design, data collection and analysis, decision to publish, or preparation of the manuscript.

**Competing interests:** The authors have declared that no competing interests exist.

are therapeutically responsive cancers that carry a 60% 5-year patient survival. However, patient survival for HPV negative (HPV-) HNSC remains lower in comparison to HPV + HNSC [2,3]. As many as 50% of HPV- HNSC patients present with advanced-stage disease associated with increased recurrence or cancer related mortality. To improve the prognosis for HPV- HNSC, more insights into the molecular processes driving its progression are required. To address this need, recent next-generation sequencing (NGS) studies have been performed for a wide range of human cancers [4]. An unexpected finding was a significant number of activating mutations in the KEAP1/NRF2 signaling pathway as well as inactivating mutations in key members of the SWI/SNF chromatin remodeling complex [4]. The fact that HPV-HNSC is commonly associated with alcohol and tobacco consumption may account for the activation of NRF2 pathway [3].

The NF-E2-related factor 2 (NFE2L2, referred to as NRF2) transcription factor induces the expression of ~200 cytoprotective genes that collectively mitigate oxidative stress and xenobiotic electrophiles, as well as reprogram metabolism [5]. Oxidative stress is a disturbance in the balance between the production and neutralization of reactive oxygen species (ROS) and reactive nitrogen species (RNS) within the cellular and tissue microenvironment. In healthy cells, ROS serve as an important molecular switch for triggering changes in signaling and metabolic pathways. However, ROS dysregulation can also result in aberrant signaling and disease, including cancer [6]. Thus, ROS levels are tightly controlled through antioxidant defenses regulated primarily by NRF2. The levels and activity of NRF2 are also closely monitored and controlled: under a quiescent state, the KEAP1/CUL3/RBX1 E3 ubiquitin ligase complex ubiquitylates NRF2 in the cytoplasm, facilitating its degradation by the 26S proteasome. In the presence of electrophiles like ROS, KEAP1 undergoes a conformational change that results in NRF2 stabilization, NRF2 nuclear translocation and transcriptional activation of NRF2 target genes [7].

The role of KEAP1-NRF2 signaling in cancer is complicated. Depending on the context and the stage of disease, NRF2 acts as either a tumor suppressive or tumor promoting factor. In normal cells lacking driver mutations in oncogenes and tumor suppressors, NRF2 provides tumor suppressive function though its ability to protect the cell from environmental toxicants such as tobacco smoke or UV radiation [8]. In cancer, elevated and constitutive NRF2 activation has been detected in HNSC patients, which correlates with poor prognosis and therapeutic resistance [9]. Prevailing models posit that proliferative and metabolic stress inherent to the cancer state creates a selective pressure for NRF2 activity. Thus, cancer cells hijack NRF2's protective functions to maintain their survival during progression. Similarly, NRF2 activity drives resistance to cytotoxic chemotherapy and radiation therapy.

The SWI/SNF complex is one of the three major chromatin remodeling complexes discovered in mammalian cells [10] The human SWI/SNF complex, a 1.5- to 2-MDa multi-subunit complex employing either BRG1 (SMARCA4) or BRM (SMARCA2) as the catalytic subunit, is comprised of three distinct subcomplexes of about 10–12 protein subunits: the canonical BAF (BAF), the polybromo-associated BAF (PBAF) and the GLTSCR1 or GLTSCR1L- and BRD9-containing (GBAF) complex, also known as non-canonical BAF (ncBAF) [11]. Each complex has a core ATPase which utilizes the energy of ATP to "slide" nucleosomes around DNA. Nucleosomes have a central role in controlling gene expression as their presence generally prevents the binding of transcription factors and RNA transcription machinery. Multiple reports establish this mechanism for how SWI/SNF exerts its regulatory role over global gene transcription [12–14]. Because of its fundamental role in regulation of gene transcription, SWI/SNF has been shown to control many critical cellular processes such as cell cycle regulation, cell differentiation and DNA repair [10,15]. Aberrant regulation of these processes can lead to many of the hallmarks of cancer cells.

SWI/SNF complexes were first implicated in driving tumorigenesis with the discovery of biallelic SMARCB1 inactivating mutations in nearly all cases of rhabdoid tumor, a cancer that typically develops in children <3 years of age with a notably poor prognosis [16]. More recently, NGS studies have found that >20% of human cancers contain a mutation in a SWI/SNF subunit [4]. ARID1A, an exclusive member of the BAF complex, was discovered to be mutated in nearly 50% of ovarian clear cell carcinomas (OCCCs) and ovarian endometrioid carcinomas (OECs) [17–19]. Similarly, mutations in PBRM1, an exclusive member of the PBAF complex, were identified in 41% of patients with clear-cell renal cell carcinoma (ccRCC) [20]. Overall, ARID1A is the most frequently mutated subunit with early reports suggesting a tumor suppressor role [21,22].

Previous reports have demonstrated a functional interaction between the SWI/SNF complex and the KEAP1/NRF2 pathway using colorectal carcinoma and immortalized embryonic kidney cell lines [23]. We previously reported that loss of the catalytic subunits of SWI/SNF (SMARCA4 and SMARCA2) in lung adenocarcinoma tumors and lung cancer cell models altered KEAP1-NRF2 signaling [24]. Given this association between the ATPase subunits and NRF2 signaling, we hypothesized that a similar relationship may exist between other frequently mutated SWI/SNF subunits and altered NRF2 signaling. Using data from The Cancer Genome Atlas (TCGA) and Clinical Proteomic Tumor Analysis Consortium (CPTAC), we explored the relationship between tumors with ARID1A mutations and NRF2 signaling because ARID1A is the most frequently mutated SWI/SNF subunit. We report that loss of ARID1A in HPV- HNSC tumors activated KEAP1-NRF2 signaling, while other tumors with frequent ARID1A mutations did not display as strong a relationship. Additionally, exploration of mutations in subunits representative of the PBAF complexes revealed variation in effects on NRF2 signaling. Our results implicate ARID1A inactivation in promoting tumor progression and survival of HPV- HNSC through activation of the KEAP1-NRF2 signaling pathway.

## Materials and methods

### TCGA data acquisition

Only de-identified, publicly available clinical and genomic data were utilized for this study. Per-gene quantified mRNA read count data, as well as per-gene discretized Gistic2 copy-number analysis data for the Cancer Genome Atlas [25]. were downloaded from the Broad Firehose Portal [26].

### CPTAC data acquisition

CPTAC data were acquired from: http://www.linkedomics.org/data_download/CPTAC-HSCC. RNA-seq, gene level somatic mutations, Proteomic, and CNV data were downloaded for each cohort and used for analysis. Samples were collected with informed consent and snap-frozen in liquid nitrogen. Tissue samples were cryopulzerized and aliquoted for DNA, RNA, or proteomic analysis. Whole exome and whole genome DNA sequencing was performed. Copy number analysis was performed using both whole-genome and whole-exome sequencing data. CNVEX pipeline (https://github.com/mctp/cnvex) was used for processing. Somatic mutations were called using Somatic sniper from Whole-exome sequencing data. RNA sequencing was performed with paired-end sequencing generating 120 million reads per library. Samples were mapped to hg38 human genome reference. Proteomics samples were prepared with trypsin digested, and TMT labeled before global proteomic analysis [27].

## Evaluating NRF2 and SWI/SNF mutational landscape

In this work, we consider a Gistic score of -2 synonymous with deep deletion, and Gistic score of -1 synonymous with a shallow deletion. Gistic uses a dynamic segmentation algorithm to define chromosomal arm level (-1) and deeper focal deletions (-2) based on per tumor thresholds [25]. Variant calls were downloaded using the R TCGAbiolinks [28] package; calls performed with VarScan [29] were used for all analyses. Any deep deletion, focal amplification, or non-synonymous variants in KEAP1, CUL3, or NFE2L2 (NRF2) were considered evidence of NRF2 pathway alteration. Any deep deletion, focal amplification, or non-synonymous variants in the SWI/SNF subunit(s) of interest (ARID1A/B, ARID2, PBRM1) was considered evidence of SWI/SNF alteration.

## Transcriptomic data pre-processing

RNA read count data was preprocessed by filtering low expression genes to obtain an approximately Gaussian distribution of $Log_2$CPM values. Filtered read count data were then normalized using the trimmed means of M values methods provided in the edgeR package [30]. The Limma-voom pipeline was used for all subsequent differential expression analysis (e.g., volcano plots, heatmaps) for methodological unity [31]. The Limma-Voom pipeline also allowed us to work with the same set of pre-processed (Normalized LCPM) expression values for both the differential expression analyses, as well as the derivation of the related centroid classifier.

## Derivation of a novel NRF2 activity signature

To construct a high-performance gene expression signature for NRF2 activity, we employed a centroid classifier, trained on high confidence class members. Classifiers were defined and cross validated using the R cancer class package [32]. Specifically, preliminary groups of NRF2 active and inactive tumors were assigned by mutational status. Specifically, all tumors with deep deletions (Gistic value = -2) in KEAP1 or CUL3, or amplifications (Gistic value = +2) for NFE2L2 or mutations (missense, nonsense, frame shift) in the KEAP1/CUL3, or missense mutations in NFE2L2 were considered NRF2 active and other tumors inactive. An initial differential expression was performed between these preliminary groups and a classifier defined based on the top 100 genes ranked by p-value. High confidence class members were defined as having correct initial assignment and having RNA expression values very similar to the class-defining average of expression (less than 0.25% of the inter-centroid distance). The gene set and classifications were then improved with a machine learning (filtering) procedure, in which tumors initially misclassified or were more than 0.25% away from a centroid were temporarily removed (filtered). Then the filtered data were then used for differential expression and construction of a final classifier. This procedure was applied individually to the TCGA HNSC, LUSC, and LUAD data sets. Genes which were found to be differentially expressed in all three cases after the machine learning step were included and prioritized by the highest of the three individually associated adjusted p-values (ensuring low values in all cases). The top 100 genes by p-value with increased expression in NRF2 active tumors were used as an NRF2 gene signature. We termed the signature the head-neck-lung (HNLU) signature.

## Gene set enrichment analysis

Ranked gene lists were created using the signal to noise ratio for the change in expression between two groups of interest as defined in the popular GSEA software package distributed by the Broad Institute [33]. Hallmark and Oncogene signatures from the MiSigDB were used as gene sets of interest [34]. GSEA testing, related multiple comparison testing, and

enrichment score normalization were performed with the R fgsea package [35]. Additional NRF2 gene signatures used to assay NRF2 transcriptional activity (Singh, Act, Slat, Onco, HNLU) are detailed in **S1 Table**.

## Results

### Mutations in SWI/SNF subunits co-occur in tumors with activating mutations in the NRF2 pathway

We previously reported that loss of SMARCA4 in lung adenocarcinomas led to increased NRF2 signaling, including expression of the NRF2 targets HMOX1 and GSTM4 [24]. To build on these observations and extend to other tumor types, we determined whether mutations in other SWI/SNF subunits correlated with activating mutations in the NRF2 pathway using data from The Cancer Genome Atlas (TCGA). First, we defined the frequency of mutations in members of the SWI/SNF complex and the NRF2/KEAP1/CUL3 complex. Grouping mutations into their respective complexes and comparison across cancer types showed that most types where $\geq 5\%$ of the tumors showed activation of the NRF2 pathway also displayed SWI/SNF mutation frequencies from 10%-50% (**Fig 1A**). To determine statistical significance, we performed a two-proportion z-test of SWI/SNF mutation frequency between tumors with $\geq 5\%$ KEAP1-NRF2-CUL3 alterations and those with $<5\%$ frequency. The test revealed tumors with $\geq 5\%$ frequencies of KEAP1-NRF2-CUL3 alterations had a statistically significant higher proportion of tumors with SWI/SNF complex mutations than those with $<5\%$ (23% vs 14%, p = 2.2 x $10^{-16}$) (**Fig 1B**). This correlation suggested that mutations in SWI/SNF subunits might promote oncogenesis in some tissues through activation and/or augmentation of NRF2 activity.

### ARID1A-mutant tumors show activation of NRF2 gene expression signatures

To further characterize the potential relationship between SWI/SNF subunit mutations and NRF2 signaling, we focused our analyses on ARID1A, the most frequently mutated SWI/SNF subunit overall (**Fig 1C**) [19]. Several recent studies have implicated a role for ARID1A loss in driving cancer progression, including HNSC [30]. Of import, Ogiwara et. al., showed that cultured and primary ovarian cancer cells lacking ARID1A expression have low SLC7A11, a downstream target of NRF2, resulting in specific vulnerability to inhibitors of the GSH metabolic pathway [36].

To investigate a potential relationship between ARID1A loss and altered NRF2 signaling, we first separated the tumors based on their mutation and CNV status and removed confounding mutations in other SWI/SNF subunits and in NRF2, KEAP1and CUL3 from each mutation group, resulting in an ARID1A mutant set (**Fig 2**). We followed a similar strategy to generate a mutant group for ARID1B, the mutually exclusive family member. We then performed signal-to-noise Gene Set Enrichment Analysis (GSEA) using four established NRF2 genes sets along with an unpublished set developed at UNC (**Fig 2**). Finally, we compared the level of enrichment for NRF2 activity between tumors containing mutations in the gene of interest and a "wild-type" groups without any known NRF2 activating mutations or SWI/SNF subunit mutations.

We first validated our approach using the KEAP1/NRF2/CUL3 mutant tumors as a positive control for increased enrichment for NRF2 signaling (**Fig 3A; S2 Table**). All tumors showed increased enrichment of the 5 NRF2 signatures except Ovarian serous adenocarcinoma (**Fig 3A**). Although stomach adenocarcinoma demonstrated negative enrichment for the Singh signature, we found only a limited number of mutations, mostly NRF2 deletions.

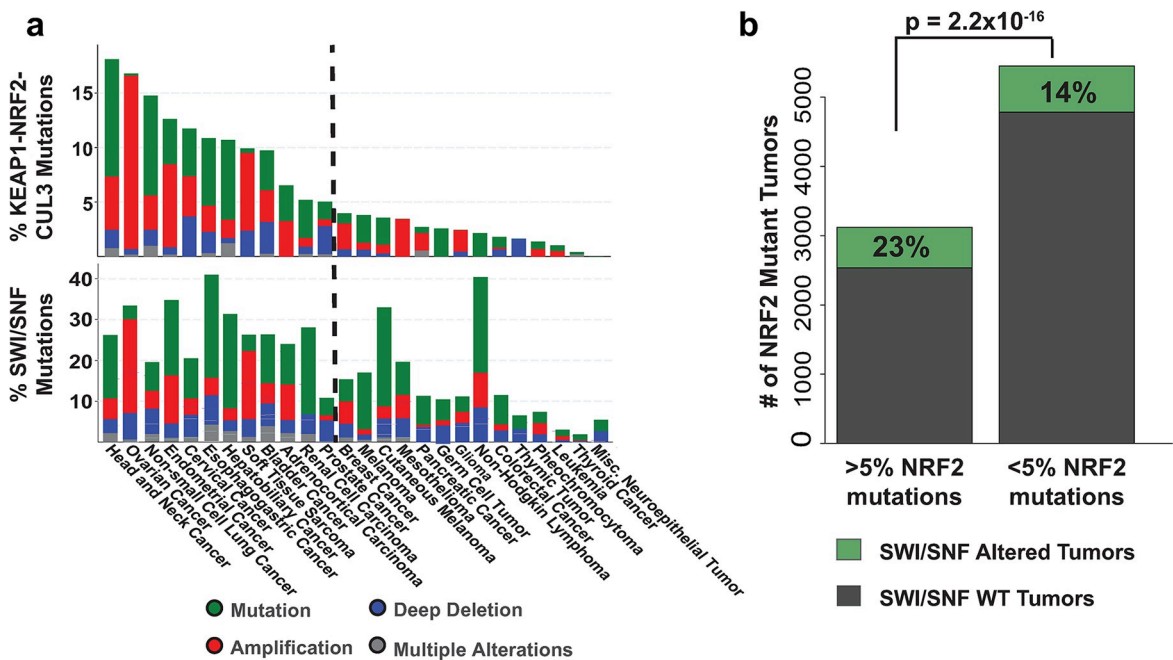

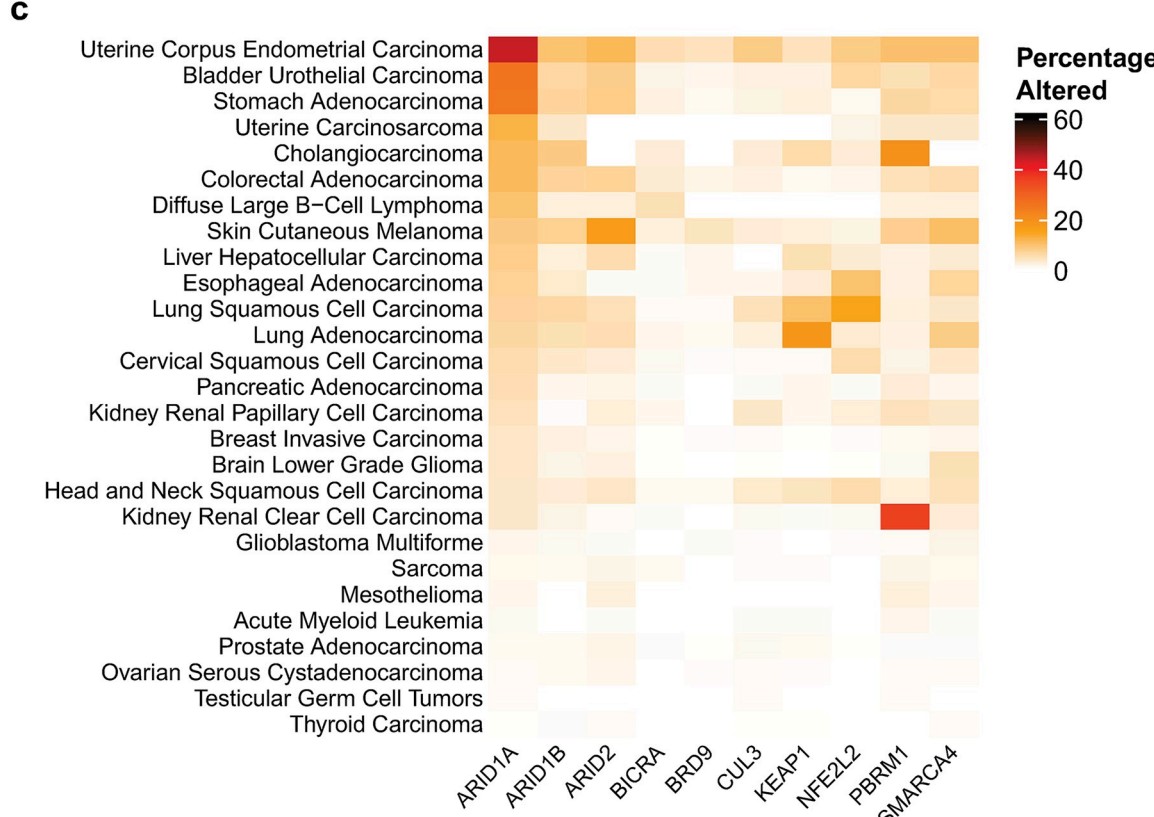

**Fig 1. SWI/SNF and KEAP1-NRF2 are frequently mutated in human cancers.** A. Frequency of coincident alterations of SWI/SNF subunits and KEAP1-NRF2-CUL3 in TCGA tumor types. B. Frequent alterations in SWI/SNF subunits in tumors with ≥ frequencies of KEAP1-NRF2 alterations. C. Heatmap demonstrating that ARID1A is the most frequently altered SWI/SNF subunit across TCGA tumors.

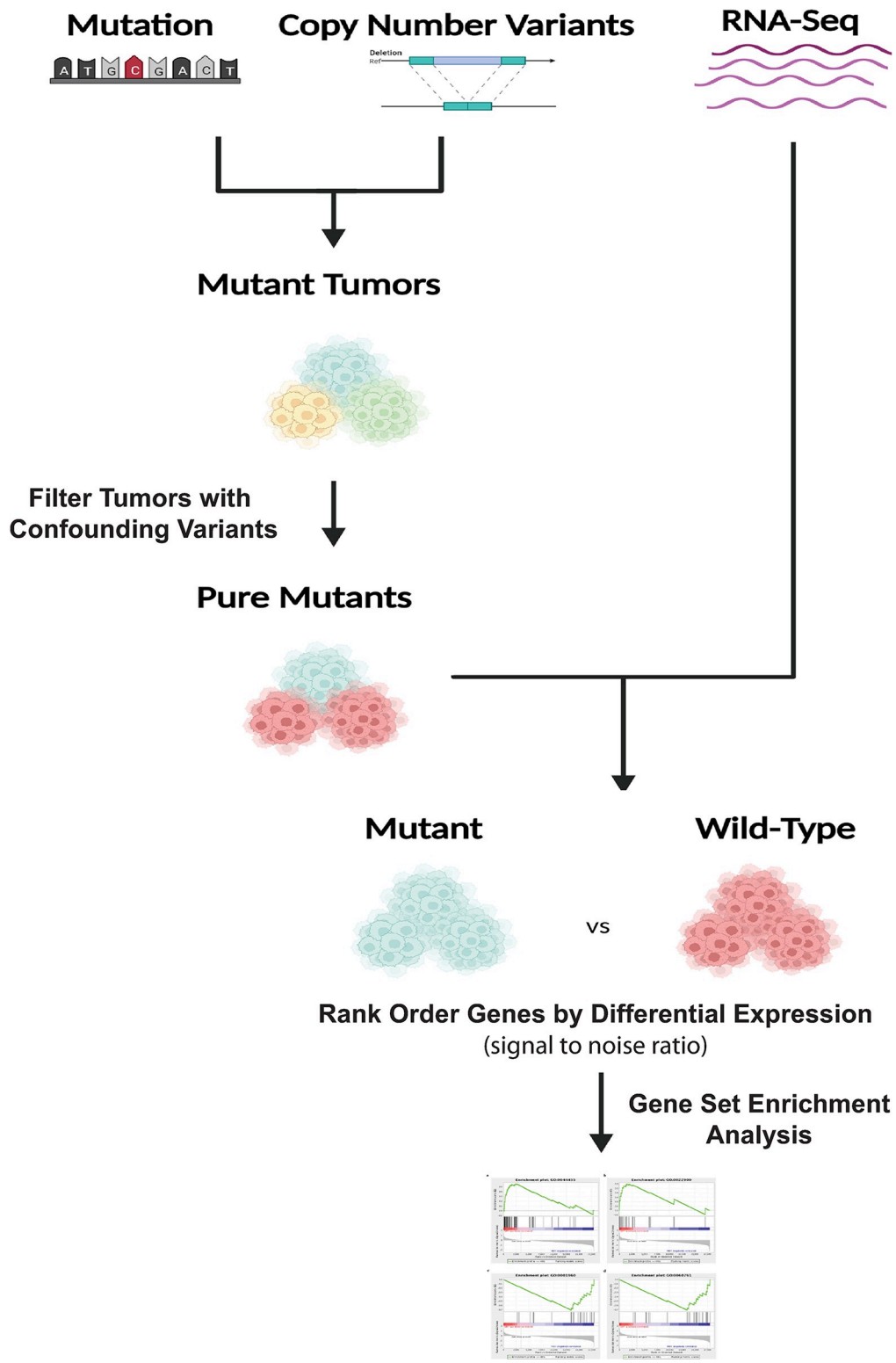

**Fig 2. TCGA bioinformatic pipeline.** TCGA Firehose was used to download CNV, mutation, and gene expression data. CNV and mutation data were used to identify mutant tumors in genes for each group. Confounding mutations were then filtered from the treatment group. Gene expression data were used to calculate signal-to-noise ratios within each group. GSEA was performed using signal-to-noise ratios between control and treated groups for NRF2 gene signatures and Oncogene and Hallmark signatures.

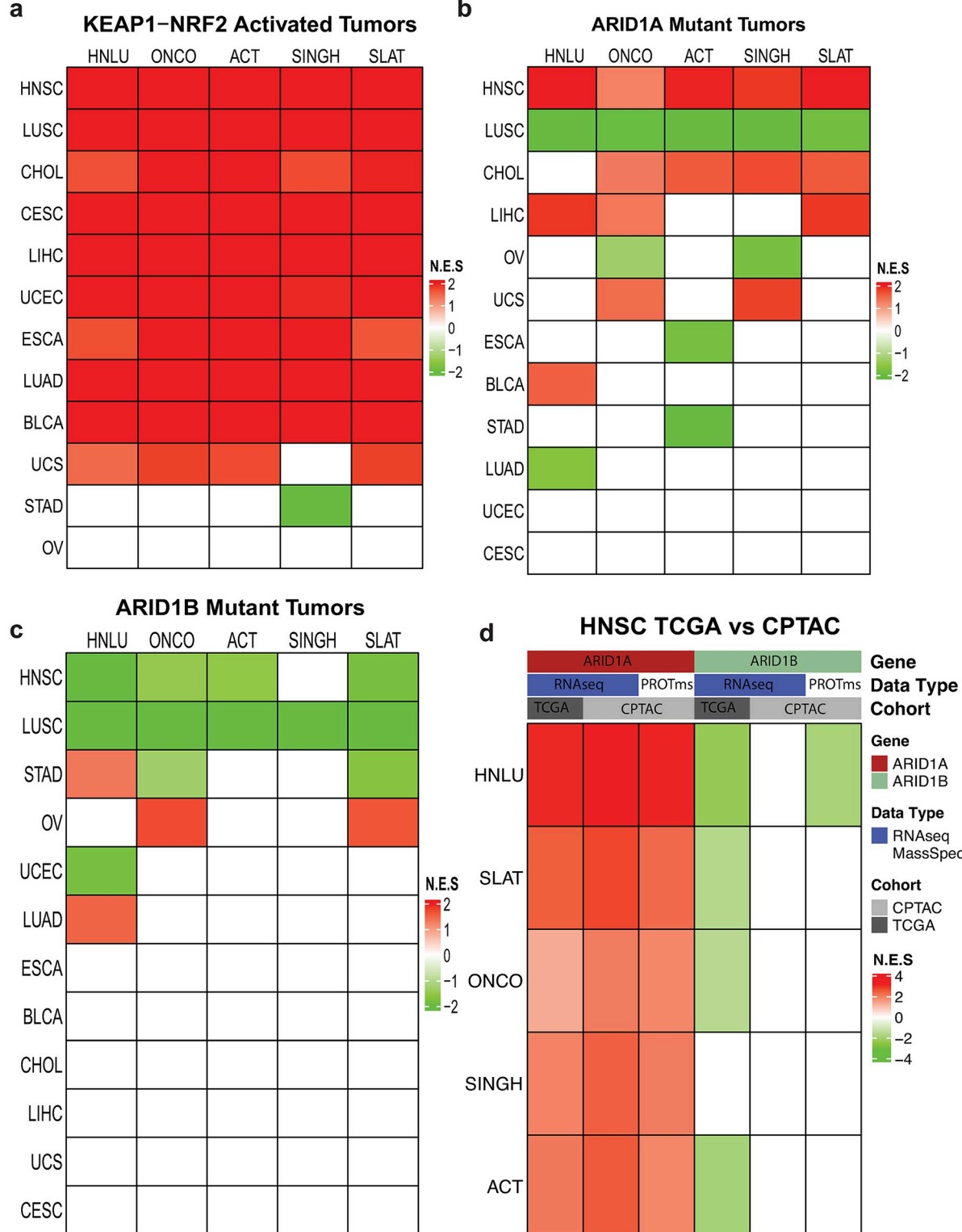

**Fig 3. ARID1A and ARID1B mutant tumors alter KEAP1-NRF2 signatures.** GSEA of five NRF2 signatures was used. Normalized Enrichment Scores (NES) are shown in the heat map with Tumors (S2 Table) on the x-axis and Signatures (S3 Table) on the y-axis. A. NRF2 signature GSEA results for KEAP1-NRF2 mutant tumors (positive control). B. NRF2 signature GSEA results for ARID1A mutant tumors C. NRF2 signature GSEA results for ARID1B mutant tumors. D. NRF2 signature GSEA results for HNSC tumors comparing the CPTAC and TCGA datasets.

We next looked at the impact of ARID1A mutations on the five NRF2 signatures. In (HNSC all five signatures were significantly increased in the presence of ARID1A mutations (**Fig 3B**). In contrast, in LUSC, the 5 signatures were decreased in the ARID1A mutant group (**Fig 3B**). Liver cancer (LIHC) and cholangiocarcinoma (CHOL) showed the next highest numbers of NRF2 signatures enriched. Of interest, tumors with higher frequencies of SWI/SNF mutations, such as stomach adenocarcinoma (STAD), bladder (BLCA) and uterine (UCEC), showed fewer changes in NRF2 signatures than the ARID1A mutant group. These tumors also had the highest ARID1A mutation rates (**S3 Table**), suggesting that higher frequencies of ARID1A mutations does not correlate with altered NRF2 signaling.

## ARID1B mutant tumors show less activation of NRF2 signaling

Some studies have shown common target genes between ARID1A and ARID1B, suggesting they define overlapping but distinct activities of the SWI/SNF complex [21]. ARID1B, the closely related subunit to ARDI1A, also only appears in the BAF complex. Therefore, we also tested whether mutations in ARID1B (**Fig 3C**) affected the activity of the NRF2 signatures. As shown in **Fig 3C**, ARID1B loss had a lesser effect on KEAP1-NRF2 signaling, as only LUSC and HNSC showed changes in enrichment of >4 NRF2 signatures. Additionally, fewer tumors demonstrated altered KEAP1-NRF2 enrichment in comparison to ARID1A (6 vs 10). ARID1B mutant LUSC demonstrated enrichment for decreased NRF2 signaling, similar to that associated with ARID1A mutations. However, ARID1B mutant HNSC showed a decrease in enrichment of four of five NRF2 signatures, the reverse of the finding for ARID1A mutant tumors (**Fig 3B and 3C**). We also attempted to define the effects of dual mutations in ARID1A and ARID1B, but the sample size proved too small for a statistically significant analysis (**S4 Table**).

## CPATC show increased NRF2 signaling in ARID1A mutant HNSC

To validate our findings in an independent dataset we queried the Clinical Proteomic Tumor Analysis Consortium (CPTAC) study as it provided RNA-seq and, importantly, proteomics data for HNSC [27]. Following the same pipeline shown in **Fig 2**, we identified tumors with ARID1A or ARID1 mutations within the CPTAC data. We observed similar results for HNSC as with the TCGA ARID1A mutations in tumors associated with an increased enrichment of the five NRF2 signatures (**Fig 3D**). In agreement with the RNA-seq results, the proteomic data from CPTAC also displayed an increase of the five NRF2 signatures (**Fig 3D**). In tumors with ARID1B mutations, none of the signatures were found significantly changed (**Fig 3D**). We did note a significant decrease in enrichment for the HNLU signature in the ARID1B mutant group for the proteomics data (**Fig 3D**).

## PBAF subunits alter KEAP1-NRF2 signaling

While ARID1A/B are exclusive subunits of the BAF complex, the ARID2 and PBRM1 subunits define the SWI/SNF complex, PBAF [37]. Because of the frequent mutations in ARID2 and PBRM1 in some cancers, we explored whether loss of either subunit altered NRF2 signaling. Following the protocol outlined in **Fig 2**, we generated ARID2 and PBRM1 mutant sets. We then analyzed the same five NRF2 gene signatures and TCGA tumor sets and found that mutations in PBAF subunits correlated with altered KEAP1-NRF2 signaling in several tumors (**Fig 4**). Mutations in ARID2 (**Fig 4A**) showed increased enrichment of four of five signatures for cholangiocarcinoma (CHOL). In contrast, loss of this subunit decreased enrichment in four signatures for HNSC and two of the five signatures for LUSC, contrasting with the ARID1A mutant results for HNSC. PBRM1 mutations in CHOL also correlated with increased enrichment in four NRF2 signatures (**Fig 4B**). PBRM1 mutations were also associated with decreased

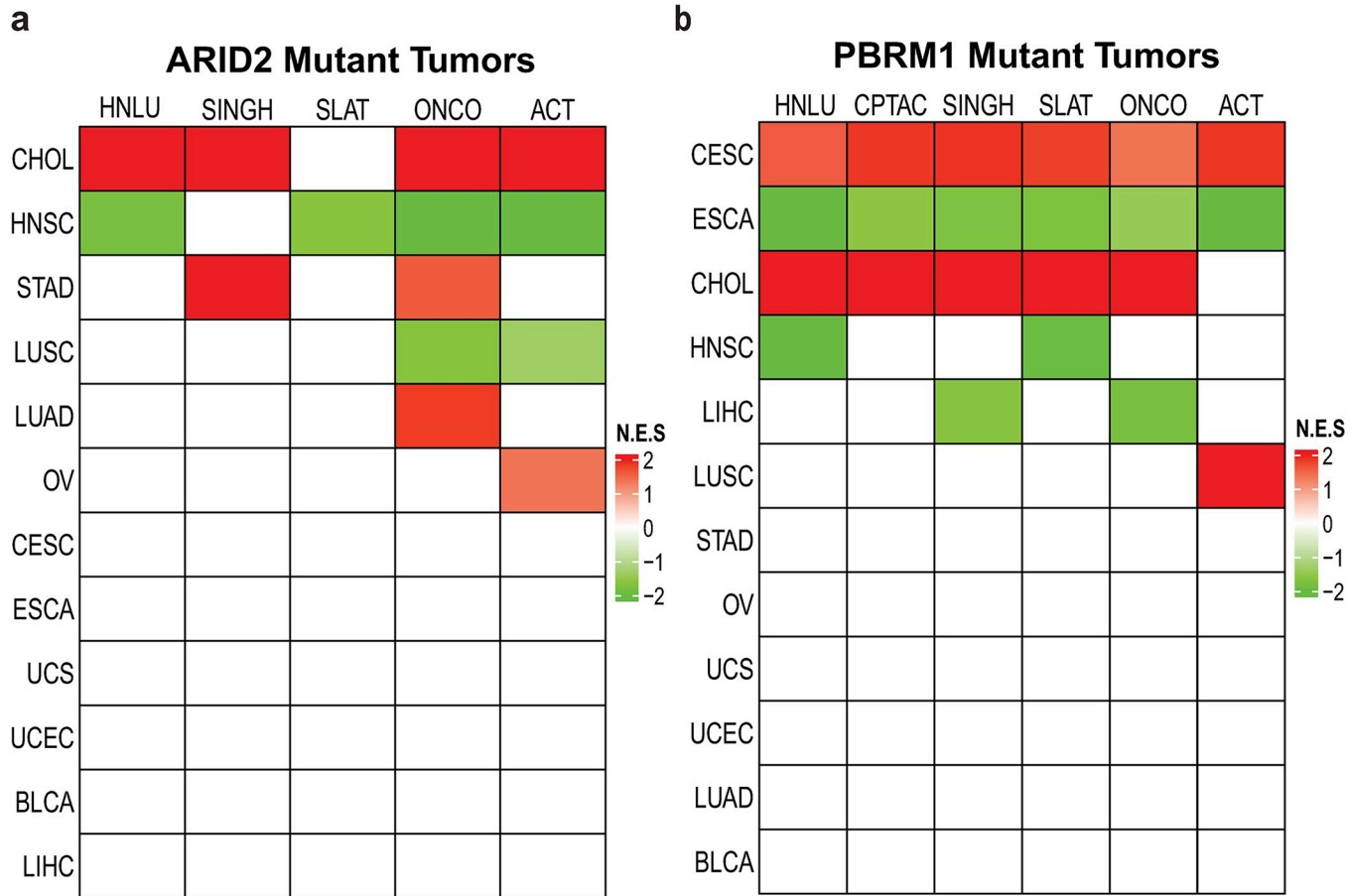

**Fig 4. ARID2 and PBRM1 mutant tumors alter KEAP1-NRF2 signatures.** GSEA of five NRF2 signatures was used. Normalized Enrichment Scores (NES) are shown in the heat map with Tumors (**S2 Table**) on the x-axis and Signatures (**S3 Table**) on the y-axis. A. NRF2 signature GSEA results for ARID2 mutant tumors. B. NRF2 signature GSEA results for PBRM1 mutant tumors.

NRF2 enrichment scores in HNSC but in only two of the signatures. Interestingly, PBRM1 mutations in cervical squamous cell carcinoma and endocervical adenocarcinoma (CESC) and esophageal carcinoma (ESCA) correlated with robust responses to all five signatures, increasing and decreasing, respectively (**Fig 4B**). This finding supports a tumor specific response to PBRM1 for these tumors as CESC only showed changes with PBRM1 mutations. Inversely, CHOL showed enrichment for increased NRF2 signaling with three of four subunit mutations, with the exception of ARID1B, suggesting a tumor-specific response as well.

## NRF2 signatures are one of the top pathways altered in the presence of SWI/SNF mutations in HNSC and LUSC

Few studies have examined the effects of SWI/SNF subunit mutations on global signaling in primary human tumors. To address whether tumors with mutations in these four subunits showed changes in enrichment in other signaling pathways, we investigated the impact of mutations in ARID1A, ARID1B, ARID2 and PBRM1 on oncogenic signaling using the Onco-gene Gene Set (**S5 Table**) and overall cellular signaling using Hallmark Signatures Set (**S6 Table**). Overall, SWI/SNF mutations in LUSC repressed global pathway enrichment in both the Oncogene and Hallmark data sets. HNSC displayed a more complicated result where BAF

complex members mostly decreased Oncogene pathway enrichment and PBAF complex members generally increased pathway enrichment. For the Hallmark signatures in HNSC, ARID1A mutations showed a neutral to positive trend while PBRM1 mutations increased enrichment of the Hallmark signatures. For LUAD, ARID1A mutations increased signaling in the Oncogene and Hallmark datasets while mutations in the other three subunits decreased signaling in both datasets. Overall, these data suggest a complicated and context-dependent impact of SWI/SNF mutations on global signaling pathways.

The data in **S5** and **S6** **Tables** show changes in enrichment of a significant number of Hallmark and Oncogene signatures in human tumors with mutations in SWI/SNF subunits. While we have also observed a strong correlation between mutations in ARID1A in HNSC and increased enrichment of NRF2 signatures, we have not established that increased NRF2 signaling represents one of the most enriched of the many altered signatures. To address this issue, we compared the normalized NRF2 enrichment scores and their significance to signatures of other canonical signaling pathways. We focused on HNSC and LUSC because they had the most robust KEAP1-NRF2 response to SWI/SNF mutations. As shown in **Fig 5A**, a volcano

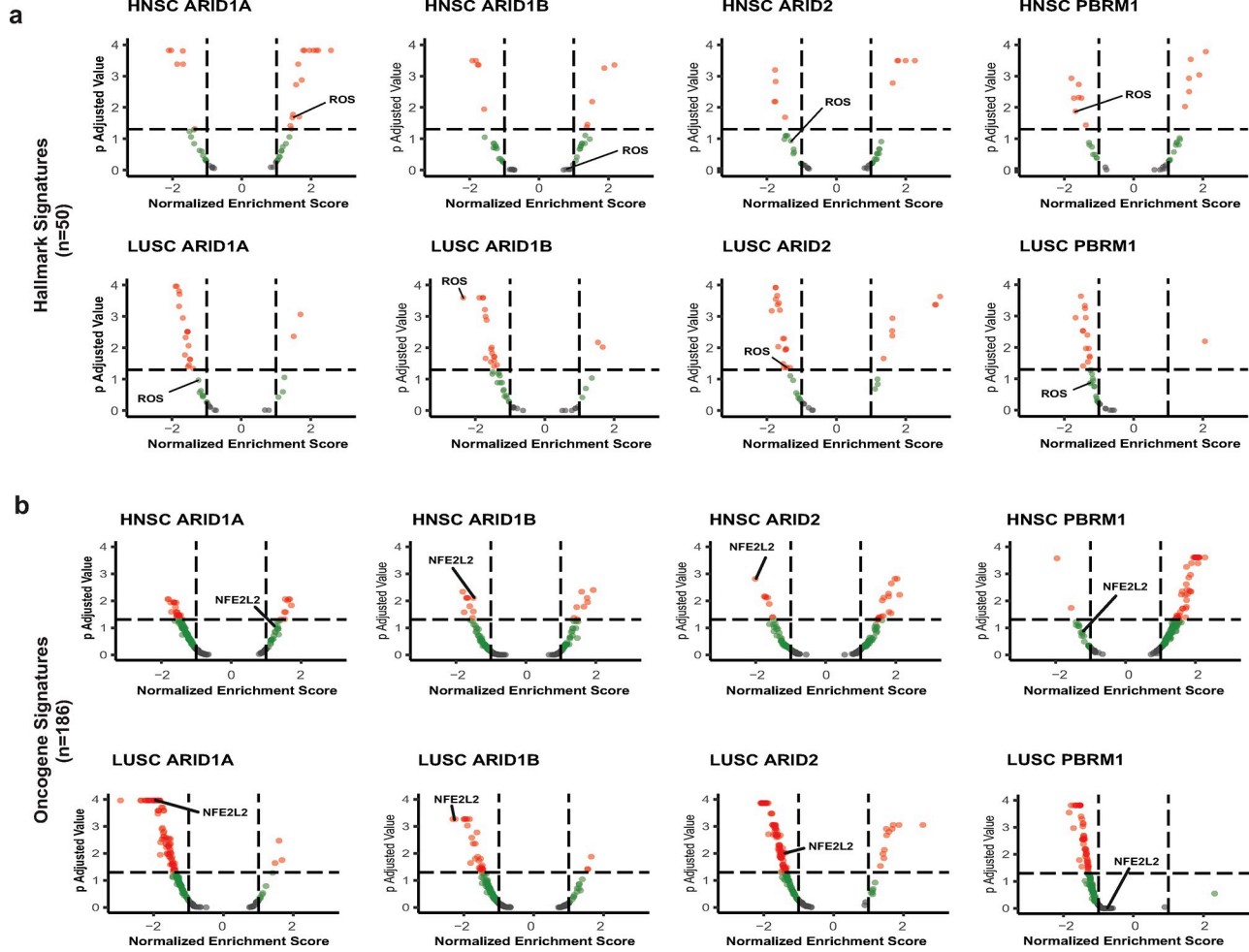

**Fig 5. NRF2 is one of the strongest pathway responses to SWI/SNF loss in hallmark and oncogene signatures.** A. Volcano plots of the canonical Hallmark Gene Set. Normalized Enrichment Scores (NES) are shown on the x-axis and p-adjusted values on the y-axis. HALLMARK_REACTIVE_OXYGEN_SPECIES (ROS) is the signature associated with NRF2 signaling. B. Volcano plots of the canonical Hallmark Gene Set. NES are shown on the x-axis and p-adjusted values on the y-axis. NFE2L2.V2 is the signature associated with NRF2 signaling.

plot analysis of the Hallmark signatures demonstrated that the NRF2 signature (HALL-MARK_REACTIVE_OXYGEN_SPECIES) was among the top alterations in terms of significance ($p < 0.05$) and NES for mutations in ARID1A and PBRM1 in HNSC. We saw similar results for LUSC where this NRF2 signature was among the top datasets altered by mutations in ARID1A, ARID1B, and ARID2 (**Fig 5A**). For the Oncogene dataset, the NRF2 signature (NFE2L2.V2) was among the top altered pathways associated with mutations in ARID1B and ARID2 in HNSC (**Fig 5B**). For LUSC, we found the NRF2 signature as one of the top affected pathways for mutations in ARID1A, ARID1B, and ARID2 (**Fig 5B**). Importantly, we observed a general decrease for the Oncogene and Hallmark datasets in LUSC for mutations in SWI/SNF subunits possessing ARID domains, pointing to an ARID domain-based mechanism for regulation of NRF2 signaling.

## Discussion

In this study, we took a bioinformatics approach to explore the connection between mutations in subunits of the SWI/SNF chromatin remodeling complex and the regulation of KEAP1-NRF2 signaling in human cancers. Most previous studies focused on loss of a specific subunit in one specific cancer type, generally using cell line models. We took advantage of recent NGS studies, such as TCGA and CPTAC, to perform a global analysis for the effects of mutations in key SWI/SNF subunits on global cellular activities and associated signaling pathways in human cancers.

Our study found that in HNSC, ARID1A mutations increased and mutations in ARID1B decreased NRF2 signaling. We saw similar results for the mutants in ARID1A in datasets from TCGA and CPTAC, providing strong support for this relationship. While we observed decreased enrichment of the NRF2 signatures changes for ARID1B mutations in HNSC, they only reached significance in the TCGA data. This may reflect the lower number of tumors in the TCGA data set versus the CPTAC (522 vs. 110). Therefore, while our findings suggest that mutations of ARID1A/B alter the NRF2 signaling in a differential manner in HNSC tumors, additional sequencing and functional data are needed for validation. However, this potential difference in their functions would be specific to HNSC because we did not observe these effects in other tumor types. Our results also support a model where ARID1A loss opens the chromatin architecture increasing NRF2's accessibility to its targets, while ARID1B loss would reduce accessibility. Based on a recent report that dual loss of ARD1A and B led to the development of HNSC in a genetically-engineered mouse model (GEMM) [38], we wanted to examine the effects of dual mutations in human HNSC but lacked sufficient samples for that analysis.

Previous studies have shown frequent activating mutations of the KEAP1-NRF2 pathway in HNSC along with increased enrichment of its signatures. Other studies have shown suggested alternative mechanisms for activated NRF2 signaling such as upregulation of the c-MYC pathway or by HPV [39,40]. To date, no report has linked mutations in SWI/SNF subunits, such as ARID1A, to activation of NRF2 signaling in HNSC. Thus, our results would be the first to implicate ARID1A mutations as another mechanism for NRF2 activation. In support of this relationship, a recent study showed worse prognosis in HNSC patients with increased NRF2 activity as well as a more aggressive phenotype in tumors with mutations in SMARCB1 and SMARCA4 [41]. However, further experiments are needed to confirm a functional interaction between activation of NRF2 signaling and mutations in SWI/SNF mutations in HNSC, such as ARID1A/B knockouts in cell culture and GEMMs. Additional functional studies could also address why ARID1A and ARID1B have differential effects on NRF2 signaling. It has been suggested ARID1A and ARID1B have competing functions [21,42]. Our data suggests this is the case in HNSC, but not in LUSC. Further evidence of clinical relevance of the

ARID1A-NRF2 axis could come from assessing outcomes for patients with activated NRF2 signaling, in the presence or absence of ARID1A mutations.

The results from the analyses of other tumors did not support a global pattern of SWI/SNF mutations on NRF2 signaling but suggest that response of NRF2 signaling to mutations in SWI/SNF subunits is tissue specific. However, we did uncover several novel relationships. We found a robust NRF2 response to PBRM1 mutations in CESC, endocervical adenocarcinoma, and ESCA. While no previous reports have suggested that PBRM1 loss plays a role in development of CESC, one report identified PBRM1 as a possible early mutation in ESCA [43]. However, this study did not include any mechanistic studies exploring the role of PBRM1 in ESCA. Of import, NRF2 activation is associated with poorer prognosis in cervical cancer patients [44] and with treatment resistance in ESCA [45]. While PBRM1 loss modulates NRF2 activity in these cancers, its loss increased NRF2 activity in CESC and decreased it in ESCA, again suggesting the impact of PBRM1 loss acts in a context specific manner. We found a single report linking PBRM1 and NRF2 activities in cancer. In the study, MUC1-C induced PBRM1 by E2F1-mediated activation at its promoters, allowing MUC1-C to form a complex with NRF2 and PBRM1 on the NRF2 target SLC7A11 to drive its transcription in human prostate cancer stem cells [46]. Further exploration of PBRM1 loss in these cancers could provide a mechanism behind its effects on NRF2 activity in these cancers. Additionally, we examined mutations in BRD9 and BIRCA, two subunits exclusive to the ncBAF complex. However, the frequency of mutations in human tumors was too limited to perform statistical analyses.

We observed a similar lack of patterns between subunits of the BAF and PBAF complexes when examining changes in the Hallmark and Oncogene signatures. This finding again may result from tissue specific differences in the activities of these two SWI/SNF complexes. LUSC did not show any changes, while LUAD had differences between ARID1A and the other subunits. This could suggest that ARID1A directs the BAF complex to distinctly different targets than the other subunits. ARID1A was also the subunit whose loss altered enrichment of NRF2 signatures in the most tumors. HNSC displayed a difference in BAF vs PBAF signaling for Oncogene signatures, but not for the Hallmark data set. This difference could possibly reflect those mutations in BAF vs PBAF subunits play divergent roles in oncogenic signaling pathways as opposed to those regulating broader signaling pathways.

Some limitations exist for this study. We attempted to confirm the LUSC results in the CPTAC, but NRF2 signature enrichment was decreased rather than the increase found with the TCGA analysis. While we considered an error in the CPATC dataset, we did find that KEAP1-NRF2 mutations in these tumors increased NRF2 signaling in both TCGA and CPTAC, as expected. In addition, the HNSC results for ARID1A mutations from the CPTAC data in the corroborated the TCGA. The CPTAC data for LUSC also did not identify the specific mutations found in each sample, in contrast to the HNSC data. Cell line experiments are needed to resolve the difference between the two data sets for LUSC. Another limitation is the inclusion of all missense mutations for each gene examined. While the functional impact for these mutations is generally known for NRF2 and KEAP1, the biological relevance in cancer for most of the missense mutations for the SWI/SNF subunits remains unknown. We searched the literature for studies identifying ARID1A missense mutations with functional relevance but could not find any. This caveat also holds true for the ARID1B mutations where most mutations were missense. Therefore, we operated under the assumption all mutations caused complete or impaired loss of function and contributed to biology of cancer.

The results of our study point to a more expansive link between altered SWI/SNF complexes and regulation of NRF2 activity than previously thought. Our study also opens new areas for further inquiry and potentially elevate SWI/SNF complexes as one of the key drivers of NRF2 activation in tumorigenesis. Because this study relied purely on bioinformatics in

terms of its approach and results, future experiments in cell culture and animal models are needed to test the biological relevance of our observations. Recent studies have shown that NRF2 activation correlates with poorer patient survival in HNSC [9]. However, the association with ARID1A loss on patient outcomes in HNSCC remains unclear with published reports showing improved survival and poorer survival [47,48]. These conflicting reports further emphasize the importance of further investigations into the functional interactions between these two oncogenic pathways.

## Conclusion

Pan-cancer analysis revealed an association between loss of SWI/SNF subunits and altered KEAP1-NRF2 signaling. The effects of SWI/SNF subunit loss on NRF2 signaling were tissue and mutation specific. ARID1A loss activated NRF2 signaling in head and neck squamous tumors, confirmed using another cancer dataset. Additionally, we report evidence of differential gene regulation between the BAF and PBAF complexes in specific cancers. We also identified several human cancers where loss of subunits of the PBAF complex correlated with activation of NRF2 signaling. Our studies identified a novel association between two disparate cellular regulatory mechanisms which may serve as a marker for personalized medicine and early detection.

## Supporting information

**S1 Table. NRF2 gene signatures used in this study.**
(XLSX)

**S2 Table. Abbreviation of TCGA tumor types.**
(XLSX)

**S3 Table. Sample sizes of treatment and control groups for each tumor type.**
(XLSX)

**S4 Table. Sample sizes of co-occurring mutant tumors.**
(XLSX)

**S5 Table. Hallmark signatures of mutant tumors.**
(XLSX)

**S6 Table. Oncogenic signatures of mutant tumors.**
(XLSX)

## Acknowledgments

We would like to thank Dr. Jesse Raab, Dr. Cavin Ward-Caviness, and Dr. Ilona Jaspers (University of North Carolina at Chapel Hill) for valuable support and input.

## Author Contributions

**Conceptualization:** Vinh Nguyen, Michael B. Major, Bernard E. Weissman.

**Data curation:** Vinh Nguyen, Travis P. Schrank.

**Formal analysis:** Vinh Nguyen, Travis P. Schrank.

**Funding acquisition:** Travis P. Schrank, Michael B. Major, Bernard E. Weissman.

**Investigation:** Michael B. Major.

**Project administration:** Michael B. Major, Bernard E. Weissman.

**Software:** Vinh Nguyen, Travis P. Schrank, Michael B. Major.

**Supervision:** Bernard E. Weissman.

**Writing – original draft:** Vinh Nguyen.

**Writing – review & editing:** Travis P. Schrank, Michael B. Major, Bernard E. Weissman.

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
