## [Decision Letter · Decision Letter 0]

21 Nov 2023

PONE-D-23-34974ARID1A loss is associated with increased NRF2 signaling in human head and neck squamous cell carcinomasPLOS ONE

Dear Dr. Weissman,

Thank you for submitting your manuscript to PLOS ONE. After careful consideration, we feel that it has merit but does not fully meet PLOS ONE’s publication criteria as it currently stands. Therefore, we invite you to submit a revised version of the manuscript that addresses the points raised during the review process.

We look forward to receiving your revised manuscript.

Kind regards,

Sudhir Kumar Rai, Ph.D

"This work was supported by the National Cancer Institute [R01 CA216051] (MBM & BEW), the National Institute of Dental and Craniofacial Research [K08 DE029241](TS) and the National Institute of Environmental Health Sciences [R01 T32ES007126] (VN)"  

Reviewers' comments:

Reviewer's Responses to Questions

**Comments to the Author**

1. Is the manuscript technically sound, and do the data support the conclusions?

Reviewer #1: Yes

Reviewer #2: Partly

2. Has the statistical analysis been performed appropriately and rigorously? 

Reviewer #1: Yes

Reviewer #2: Yes

3. Have the authors made all data underlying the findings in their manuscript fully available?

Reviewer #1: Yes

Reviewer #2: Yes

4. Is the manuscript presented in an intelligible fashion and written in standard English?

Reviewer #1: Yes

Reviewer #2: Yes

5. Review Comments to the Author

Reviewer #1: Title: ARID1A loss is associated with increased NRF2 signaling in human head and neck

squamous cell carcinomas

This is a very nicely performed study. The authors demonstrated how NRF2 and SWI/SNF mutations are related to human malignancies, and they suggested that ARID1A inactivation in HPV-negative HNSC promotes tumour growth and survival by activating the KEAP1-NRF2 signalling pathway. Overall, this is a quite solid paper that provides clear evidence.

I have a few minor comments and hope they help the authors.

1. The full stop is missing in many sentences. Please correct this throughout the manuscript.

2. The Limma-voom pipeline was used for all subsequent differential expression analysis(31)'. Why did the authors choose the Limma loop pipeline in place of DeSeq as DeSeq finds more differentially expressed genes?

Reviewer #2: In this study, author explored the relationship between mutations in BAF and PBAF subunits of the SWI/SNF complex and activation of NRF2 signal transduction by open source data. However, these conclusions only supported by data of DNA and RNA level. We did not find out the mechanism or clinical significance from this paper.

6. PLOS authors have the option to publish the peer review history of their article (what does this mean?). If published, this will include your full peer review and any attached files.

Reviewer #1: No

Reviewer #2: No

---

## [Author Response · Author response to Decision Letter 0]

5 Jan 2024

The manuscript was edited by the manuscript preparation expert in our Cancer Center. She found multiple errors and corrected them. We apologize for not catching them before the initial submission.

2. Please note that PLOS ONE has specific guidelines on code sharing for submissions in which author-generated code underpins the findings in the manuscript. In these cases, all author-generated code must be made available without restrictions upon publication of the work. 

We acknowledge and agree with this policy.

We have corrected this issue.

"This work was supported by the National Cancer Institute [R01 CA216051] (MBM & BEW), the National Institute of Dental and Craniofacial Research [K08 DE029241] (TS) and the National Institute of Environmental Health Sciences [T32 ES007126] (VN)" 

The statement “The funders had no role in study design, data collection and analysis, decision to publish, or preparation of the manuscript.” is correct. We have added this statement to the Cover Letter.

Reviewers’ comments:

Reviewer #1: Title: ARID1A loss is associated with increased NRF2 signaling in human head and neck

squamous cell carcinomas.

This is a very nicely performed study. The authors demonstrated how NRF2 and SWI/SNF mutations are related to human malignancies, and they suggested that ARID1A inactivation in HPV-negative HNSC promotes tumour growth and survival by activating the KEAP1-NRF2 signalling pathway. Overall, this is a quite solid paper that provides clear evidence.

I have a few minor comments and hope they help the authors.

1. The full stop is missing in many sentences. Please correct this throughout the manuscript.

We again apologize for the grammatical errors. We have corrected them in the revised manuscript.

2. The Limma-voom pipeline was used for all subsequent differential expression analysis(31)'. Why did the authors choose the Limma loop pipeline in place of DeSeq as DeSeq finds more differentially expressed genes? 

Deseq2 differential expression function accepts raw count matrices. However, we desired to perform differential expression followed by defining classifiers based on the nearest centroid method. For centroids to be reasonable, statistical constructs library size normalization is needed. Limma-Voom allowed us to work with the same set of pre-processed (Normalized LCPM) expression values for both the differential expression, as well as the related centroid classifier. This approach seemed ideal as the differential expression step principally suggested genes/features that would perform well as the basis for a classifier. With this initial motivation, all other differential expression-based analyses (e.g., volcano plots, heatmaps) in the paper were performed with Limma-Voom for methodological unity.

Reviewer #2: In this study, author explored the relationship between mutations in BAF and PBAF subunits of the SWI/SNF complex and activation of NRF2 signal transduction by open source data. However, these conclusions only supported by data of DNA and RNA level. We did not find out the mechanism or clinical significance from this paper.

This study was designed as a bioinformatic query of the potential relationship between loss of ARID1A expression and activation of NRF2 signaling. The results from this manuscript lay the groundwork for the more extensive in vitro and in vivo biological and molecular studies required to identify underlying mechanisms. 

We also added a new paragraph at the end of the Discussion to emphasize the clinical significance of NRF2 activation and ARID1A in patients with HNSC.

---

## [Editor Report · Decision Letter 1]

12 Jan 2024

ARID1A loss is associated with increased NRF2 signaling in human head and neck squamous cell carcinomas

PONE-D-23-34974R1

Dear Dr.Bernard Weissman,

We’re pleased to inform you that your manuscript has been judged scientifically suitable for publication and will be formally accepted for publication once it meets all outstanding technical requirements.

Kind regards,

Sudhir Kumar Rai, Ph.D

Academic Editor

PLOS ONE

---

## [Editor Report · Acceptance letter]

6 Feb 2024

PONE-D-23-34974R1 

PLOS ONE

Dear Dr. Weissman, 

I'm pleased to inform you that your manuscript has been deemed suitable for publication in PLOS ONE. Congratulations! Your manuscript is now being handed over to our production team.

Kind regards, 

on behalf of

Dr. Sudhir Kumar Rai 

Academic Editor

PLOS ONE